# Lipschitz-Certifiable Training with a Tight Outer Bound

**Sungyoon Lee**
Seoul National University
Seoul, Korea
goman1934@snu.ac.kr

**Jaewook Lee**
Seoul National University
Seoul, Korea
jaewook@snu.ac.kr

**Saerom Park**
Sungshin Women's University
Seoul, Korea
psr6275@sungshin.ac.kr

## Abstract

Verifiable training is a promising research direction for training a robust network. However, most verifiable training methods are slow or lack scalability. In this study, we propose a fast and scalable certifiable training algorithm based on Lipschitz analysis and interval arithmetic. Our certifiable training algorithm provides a tight propagated outer bound by introducing the box constraint propagation (BCP), and it efficiently computes the worst logit over the outer bound. In the experiments, we show that BCP achieves a tighter outer bound than the global Lipschitz-based outer bound. Moreover, our certifiable training algorithm is over 12 times faster than the state-of-the-art dual relaxation-based method; however, it achieves comparable or better verification performance, improving natural accuracy. Our fast certifiable training algorithm with the tight outer bound can scale to Tiny ImageNet with verification accuracy of 20.1% ($\ell_2$-perturbation of $\epsilon = 36/255$). Our code is available at https://github.com/sungyoon-lee/bcp.

## 1 Introduction

Deep learning has shown successful results in many applications. However, it has been demonstrated that deep neural networks are vulnerable to small but adversarially designed perturbations in the input which can mislead a network to predict a wrong label [33]. There have been many studies on such adversarial attacks and defenses against them [12, 19, 28, 27, 24, 4, 35, 41, 14].

However, Athalye et al. [1] have shown that many of these defense methods are designed to defend against specific predefined adversarial attacks, and, in turn, the models can yet be broken by unseen stronger adaptive adversaries. Thus, many verification methods are proposed to guarantee stable prediction of input within a perturbation set [18, 5, 9, 16, 38, 34, 23, 11, 3, 30, 8, 43, 2]. Verification of a neural network provides either lower bounds on the norm of the input perturbations required to fool the network or upper bounds on the worst-case errors of the network against specified perturbations. In particular, verifiable training incorporates the verification procedure using the upper bound into the training loop and yields a robust model [39, 40, 7, 8, 29, 30].

Verifiable training methods are mainly categorized into two approaches: dual relaxation and layer-wise bound propagation approaches. The dual relaxation approach formulates the verifiable training as a convex optimization and uses duality to build a relaxed bound of the optimization problem and to relieve the computational load [39, 40, 29, 7, 30]. Although these verifiable training methods can provide relatively exact robustness bounds for verification, they still involve expensive computations and poor scalability. In contrast, the layer-wise bound propagation approach calculates the upper bounds on the worst case error through relaxation on the layer-wise operations and forward propagation for the perturbation set that can be made of $\ell_\infty$- or $\ell_2$-balls [13, 44, 36, 25, 32, 37]. These layer-wise methods are computationally efficient but have loose bounds in the initial phase of training, hindering the application to larger networks.

Apart from these deterministic verification approaches, randomized smoothing is one promising approach to procure robust classification results. This can make any classifier to acquire a certified adversarial robustness by constructing a smoothed classifier [22, 21, 6, 31]. However, the randomized smoothing methods require a large number of samples to certify the classifier.

In this study, we propose an efficient certifiable training method with a tight outer bound propagation. This propagation enables the model to scale to Tiny ImageNet. Our algorithm minimizes an upper bound on the robust classification error. We further tighten the upper bound by introducing a valid box constraint into the optimization problem, thereby improving the optimal solution. By tightening the upper bound on the objective, both the robustness and standard accuracy of our method improve.

To summarize, the main contributions of this paper are as follows:

- We propose a fast certifiable training algorithm called Box Constraints Propagation (BCP) with an efficient computation of the upper bound on the robust classification error. BCP is over 12 times faster than the state-of-the-art dual relaxation-based method [40].

- We can obtain tighter outer bounds than those without BCP. These bounds are on average 25.3-55.4% tighter in terms of the length of the worst logit translation. Therefore, our certificate using BCP achieves the verification accuracy comparable to CAP [40], while improving the natural accuracy on CIFAR-10.

- Our approach can scale to Tiny ImageNet and learn a certificate that can achieve 20.1% verification accuracy ($\epsilon = 36/255$). To the best of our knowledge, this is the first non-trivial (deterministic) verification accuracy on Tiny ImageNet under the $\ell_2$-robustness.

- Our verification loss can adapt to the input locations; thus, the model can learn a behavior depending on the input locations.

## 2 Certifiable Training with Worst Logit

In this section, we introduce the robust training problem for multi-class classification, define the specification based on the worst logit, and establish the objective that provides an upper bound of the robust training problem.

**Notation** We consider a $c$-class classification problem, where $\boldsymbol{x} \in \mathcal{X} \subset \mathbb{R}^N$ is an input, $y \in \mathcal{Y} = \{0, 1, ..., c-1\}$ is the label with respect to the input $\boldsymbol{x}$, and $c$ is the number of classes. A mapping that takes an input $\boldsymbol{x}$ and outputs a logit vector $\boldsymbol{\zeta} = z(\boldsymbol{x}) \in \mathcal{Z}$ is denoted by $z : \mathcal{X} \to \mathcal{Z} \subset \mathbb{R}^c$, and the corresponding classifier is $f : \mathcal{X} \to \mathcal{Y}$ with $f(\boldsymbol{x}) = \text{argmax}_{m \in \mathcal{Y}} z_m(\boldsymbol{x})$ where $z_m$ is the output logit for a class $m \in \mathcal{Y}$. We assume the classifier network is a feedforward network with $K$ layers as $\mathbf{z}^{(k)} = h^{(k)}(\mathbf{z}^{(k-1)})$, $k = 1, \ldots, K$, where $\mathbf{z}^{(k)}$ is the vector of the activations in the $k$-th layer, $\mathbf{z}^{(K)} = \boldsymbol{\zeta}$, $\mathbf{z}^{(0)} = \boldsymbol{x}$, and $h^{(k)}$ is the operation in the $k$-th layer. Let $\mathbb{B}(\boldsymbol{x}, \epsilon)$ be a perturbation set around the input $\boldsymbol{x}$ with a level of perturbation $\epsilon$. Then, for a classifier $f$, the robust classification error within the perturbation set $\mathbb{B}(\cdot, \epsilon)$ on a data distribution $\mathcal{D}$ is defined as $R(f) = \mathbb{P}_{\mathcal{D}}\big[\exists \boldsymbol{x}' \in \mathbb{B}(\boldsymbol{x}, \epsilon) \text{ s.t. } f(\boldsymbol{x}') \neq y\big]$. We omit the dependency on $\mathcal{D}$ and $\epsilon$ for simplicity.

### 2.1 Robust Training

The main goal of certifiable training is to minimize the robust classification error $R(f)$. However, because the exact verification for $R(f)$ is NP-complete [18], a simple surrogate of $R(f)$ is used to construct the objective of certifiable training. Our certifiable training minimizes an upper bound on $R(f)$ that builds a certificate of robustness whereas adversarial training [24] minimizes a lower bound on $R(f)$. To obtain the upper bound on $R(f)$, we propagate the perturbation set $\mathbb{B}(\boldsymbol{x}, \epsilon)$ and calculate the outer bound on the propagated image in the logit space $\mathcal{Z}$. For simplicity, $\mathbb{B}(\boldsymbol{x})$ denotes the input perturbation set $\mathbb{B}(\boldsymbol{x}, \epsilon)$. Let $\hat{z}(\mathbb{B}(\boldsymbol{x})) \subset \mathcal{Z}$ be an outer bound on the logit image of the perturbation set $z(\mathbb{B}(\boldsymbol{x}))$. Then, we can construct the following upper bound $\hat{R}(f)$ on $R(f)$:

$$
\begin{aligned}
R(f) &= \mathbb{E}_{(\boldsymbol{x},y)\sim\mathcal{D}}\Big[ \max_{\boldsymbol{\zeta} \in z(\mathbb{B}(\boldsymbol{x}))} \max_{y' \neq y} \mathbf{1}\big[(\zeta_y - \zeta_{y'}) \leq 0\big]\Big] \\
&\leq \mathbb{E}_{(\boldsymbol{x},y)\sim\mathcal{D}}\Big[ \max_{\boldsymbol{\zeta} \in \hat{z}(\mathbb{B}(\boldsymbol{x}))} \max_{y' \neq y} \mathbf{1}\big[(\zeta_y - \zeta_{y'}) \leq 0\big]\Big] = \hat{R}(f),
\end{aligned}
\tag{1}
$$

where $\zeta_m$ is the $m$-th element of the logit vector $\boldsymbol{\zeta}$ and $\mathbf{1}[\cdot]$ denotes the indicator function.

## 2.2 Worst-Translated Logit

Based on the upper bound $\hat{R}(f)$ in (1), we can construct an objective for verifiable training as $\mathbb{E}_{(\boldsymbol{x},y)\sim\mathcal{D}}\big[\max_{\boldsymbol{\zeta}\in\hat{z}(\mathbb{B}(\boldsymbol{x}))}\mathcal{L}(\boldsymbol{\zeta},y)\big]$ using cross-entropy loss $\mathcal{L}$ as a surrogate loss function for the 0-1 loss of $\hat{R}(f)$. However, it is still inefficient to find the optimal solution for the non-convex maximization problem $\max_{\boldsymbol{\zeta}\in\hat{z}(\mathbb{B}(\boldsymbol{x}))}\mathcal{L}(\boldsymbol{\zeta},y)$. Dual relaxation approach addressed this problem by computing a differentiable upper bound on the robust classification error, using a feasible dual solution of the underlying relaxed LP [39, 40, 8]. In contrast, layer-wise propagation approach proposed the worst-case logit or the certifiable margin in the logit space to obtain a differentiable upper bound on the robust classification error [36, 13, 44]. In this study, we introduce the worst-translated logit $\underline{z}(\boldsymbol{x})$ that provides an upper bound on the cross-entropy loss over an outer bound $\hat{z}(\mathbb{B}(\boldsymbol{x}))$ as follows:

**Definition 1.** *The worst-translated logit over an outer bound $\hat{z}(\mathbb{B}(\boldsymbol{x}))$ for the input $\boldsymbol{x}$ and the corresponding label $y$ is defined as $\underline{z}(\boldsymbol{x};y) = z(\boldsymbol{x}) + t(\boldsymbol{x};y)$ where the translation vector $t(\boldsymbol{x};y)$ has its $m$-th element with*

$$t_m(\boldsymbol{x};y) = (z_y(\boldsymbol{x}) - z_m(\boldsymbol{x})) - \min_{\boldsymbol{\zeta}\in\hat{z}(\mathbb{B}(\boldsymbol{x}))}\big(\zeta_y - \zeta_m\big). \tag{2}$$

*When the context is clear, we omit $y$ in $\underline{z}(\boldsymbol{x};y)$ and $t(\boldsymbol{x};y)$, and just write $\underline{z}(\boldsymbol{x})$ and $t(\boldsymbol{x})$ for brevity.*

**Proposition 1** (Wong and Kolter [39]). *For an outer bound $\hat{z}(\mathbb{B}(\boldsymbol{x})) \supset z(\mathbb{B}(\boldsymbol{x}))$ and its corresponding worst-translated logit $\underline{z}(\boldsymbol{x})$, the following inequality holds:*

$$\max_{\boldsymbol{\zeta}\in\hat{z}(\mathbb{B}(\boldsymbol{x}))}\mathcal{L}(\boldsymbol{\zeta},y) \leq \mathcal{L}(\underline{z}(\boldsymbol{x}),y), \tag{3}$$

*where $\mathcal{L}$ is the cross-entropy loss function.*

Finally, the objective to be minimized is formulated as follows:

$$\mathcal{J}(f,\mathcal{D}) = \mathbb{E}_{(\boldsymbol{x},y)\sim\mathcal{D}}\big[\mathcal{L}(\underline{z}(\boldsymbol{x}),y)\big]. \tag{4}$$

Note that the worst-translated logit $\underline{z}(\boldsymbol{x})$ for the input $\boldsymbol{x}$ may not be inside the outer bound $\hat{z}(\mathbb{B}(\boldsymbol{x}))$. The remaining problem is how to calculate the outer bound $\hat{z}(\mathbb{B}(\boldsymbol{x}))$ of the logit image $z(\mathbb{B}(\boldsymbol{x}))$ and how to solve the minimization in (2) corresponding to the outer bound, which will be discussed in Section 3.1 and 3.2, respectively.

Furthermore, by using the worst-translated logit $\underline{z}(\boldsymbol{x})$ as a certificate that guarantees robustness to adversarial perturbations, we can obtain verification error of the model $f$ on the test data $\mathcal{D}_{test}$ as follows:

$$R_V(f) = \hat{\mathbb{P}}_{(\boldsymbol{x},y)\sim\mathcal{D}_{test}}\big[\min_{y'\neq y}\big(\underline{z}_y(\boldsymbol{x}) - \underline{z}_{y'}(\boldsymbol{x})\big) \leq 0\big] \tag{5}$$

which is larger than the robust classification error $R(f)$ on the test data $\mathcal{D}_{test}$.

# 3 Lipschitz-Certifiable Training with Tight Outer Bound

In this section, we propose a tight outer bound estimation and an efficient algorithm for calculating the worst-translated logit. We mainly focus on $\ell_2$-perturbation sets in the input space, but our method can be easily extended to any $\ell_p$-perturbations for $p > 0$ and $\ell_\infty$-perturbations, as described later.

**Notation** The $\ell_2$-perturbation set and the $\ell_\infty$-perturbation set in the input space are denoted by $\mathbb{B}_2(\boldsymbol{x},\epsilon) = \{\boldsymbol{x}' : \|\boldsymbol{x}' - \boldsymbol{x}\|_2 \leq \epsilon\}$ and $\mathbb{B}_\infty(\boldsymbol{x},\epsilon) = \{\boldsymbol{x}' : |x_i' - x_i| \leq \epsilon, \forall i\}$, respectively. To obtain a tight outer bound $\hat{z}(\mathbb{B}_2(\boldsymbol{x},\epsilon))$, we propagate the perturbation sets through the layers and calculate layerwise outer bounds $\mathbb{B}_2^{(k)}$ and $\mathbb{B}_\infty^{(k)}$ in the $k$-th layer. The $k$-th layer $\ell_\infty$-bound $\mathbb{B}_\infty^{(k)}$ can be represented as the box constraint $\mathbb{B}_\infty^{(k)} = \text{midrad}(\boldsymbol{m}^{(k)}, \boldsymbol{r}^{(k)}) \equiv \{\boldsymbol{p} : |p_i - m_i^{(k)}| \leq r_i^{(k)}, \forall i\}$ with the midpoint $\boldsymbol{m}^{(k)}$ and the radius $\boldsymbol{r}^{(k)}$ [26]. We call the $\mathbb{B}_2^{(k)}$ "ball outer bounds" and the $\mathbb{B}_\infty^{(k)}$ "box outer bounds".

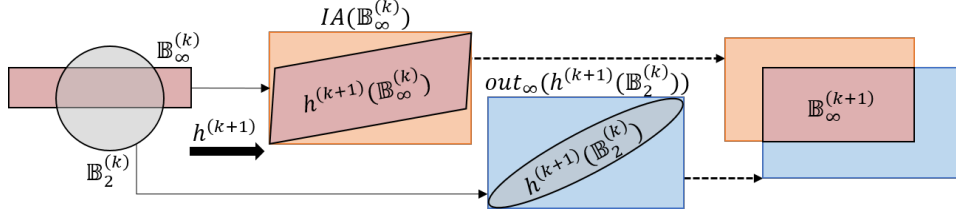

Figure 1: Illustration of BCP. For the $k$-th layer, the $k$-th box $\mathbb{B}_\infty^{(k)}$(left) is propagated to the next box $\mathbb{B}_\infty^{(k+1)}$(right), both colored in red. Note that the $k$-th ball $\mathbb{B}_2^{(k)}$ is independently propagated to the next ball $\mathbb{B}_2^{(k+1)}$ which has $L^{(k)}$ times larger radius.

## 3.1 Outer Bound Estimation

The outer bound from $\ell_2$-perturbation $\hat{z}(\mathbb{B}_2(\boldsymbol{x}, \epsilon))$ can be simply constructed by using the global Lischiptz constant $L$ of the logit function $z$, where $\hat{z}(\mathbb{B}_2(\boldsymbol{x}, \epsilon)) = \mathbb{B}_2(z(\boldsymbol{x}), \epsilon L)$ [36]. The global Lipschitz constant is efficiently computed as the product of all layer-wise Lipschitz constants, $L = \prod_{k=1}^{K} L^{(k)}$. To tighten the spherical outer bound in the logit space, Tsuzuku et al. [36] replaced it with the ellipsoidal outer bound, $\hat{z}(\mathbb{B}_2(\boldsymbol{x}, \epsilon)) = h^{(K)}(\mathbb{B}_2(\mathbf{z}^{(K-1)}, \rho^{(K-1)}))$, where $\rho^{(k)} = \epsilon \prod_{i=1}^{k} L^{(i)}$. In the objective (4), the ellipsoidal outer bound enables the Lipschitz-margin to solve the optimization in (2) explicitly. However, the global Lipschitz constant can still overestimate the outer bound and impose a strong penalty for it, limiting the expressiveness of the model [17]. It leads to a poor classification performance, not getting sharp transitions near decision boundary [15]. On the other hand, using local Lipschitz constants to estimate the outer bounds for each given input $\boldsymbol{x}$ is computationally infeasible to be integrated into the training loop for medium-sized networks, and thus limited to 2-layered networks [15, 10, 20].

To this end, we propose a method called BCP, using layer-wise propagation with the Lipschitz constant and interval arithmetic to efficiently approximate the propagation of the perturbation set adaptive to the input location. This addresses the problems of the global Lipschitz constant-based outer bound and remains the efficient computations for certifiable training. In addition, it enables us to obtain a certificate that can adapt to local properties of the classifier. We further discussed the intuition behind the design of BCP in the supplementary material.

**Box Constraint Propagation**   Our outer bound propagation starts with the $\ell_2$- and $\ell_\infty$- perturbations sets $(\mathbb{B}_2^{(0)}, \mathbb{B}_\infty^{(0)})$, propagates them through layers, and derive the tight $\ell_\infty$-outer bound $\mathbb{B}_\infty^{(k)}$ in each layer by circumscribing the propagated images and finding the intersection of the circumscribed boxes. In the penultimate layer, we combine the propagated box constraint bound $\mathbb{B}_\infty^{(K-1)}$ and the propagated global Lipschitz bound $\mathbb{B}_2^{(K-1)}$ to tighten the final outer bound $\hat{z}(\mathbb{B}(\boldsymbol{x}))$.

For $\ell_2$-certifiable training, we first consider the pair $(\mathbb{B}_2^{(0)}, \mathbb{B}_\infty^{(0)})$, where $\mathbb{B}_2^{(0)} \equiv \mathbb{B}_2(\boldsymbol{x}, \epsilon)$ and $\mathbb{B}_\infty^{(0)} \equiv \mathbb{B}_\infty(\boldsymbol{x}, \epsilon)$ circumscribing $\mathbb{B}_2^{(0)}$ in the input space. Next, we propagate them through the layers to compute the layerwise outer bound pair $(\mathbb{B}_2^{(k)}, \mathbb{B}_\infty^{(k)})$. Here, we assume a feedforward network, but we can extend it to residual networks (see the supplementary material). The ball outer bound in the $k$-th layer $\mathbb{B}_2^{(k)}$ is the Lipschitz outer bound $\mathbb{B}_2(\mathbf{z}^{(k)}, \rho^{(k)})$ with the radius $\rho^{(k)} = \epsilon \prod_{i=1}^{k} L^{(i)}$, where we use the power iteration to estimate the layer-wise Lipschitz constants $L^{(i)}$. The box outer bound $\mathbb{B}_\infty^{(k+1)}$ in the $(k + 1)$-th layer ($k = 0, 1, \cdots, K - 2$) is obtained by two box constraints that one circumscribes the propagated ellipse image $h^{(k+1)}(\mathbb{B}_2^{(k)})$ of the ball $\mathbb{B}_2^{(k)}$ and the other circumscribes the propagated parallelepiped image $h^{(k+1)}(\mathbb{B}_\infty^{(k)})$ of the box $\mathbb{B}_\infty^{(k)}$ as described in Figure 1.

In case of linear layers, the circumscribed box about the propagated ellipse image, $out_\infty(h^{(k+1)}(\mathbb{B}_2^{(k)}))$, is calculated as follows:

$$out_\infty(h^{(k+1)}(\mathbb{B}_2^{(k)})) = \mathrm{midrad}(\hat{\boldsymbol{m}}^{(k)}, \hat{\boldsymbol{r}}^{(k)}) \text{ s.t. } \hat{\boldsymbol{m}}^{(k)} = h^{(k+1)}(\mathbf{z}^{(k)}), \ \hat{r}_i^{(k)} = \|\mathbf{W}_{i,:}^{(k+1)}\|_2 \, \rho^{(k)},$$
(6)

where $\mathbf{W}_{i,:}^{(k+1)}$ is the $i$-th row of the weight matrix $\mathbf{W}^{(k+1)}$ of the linear function $h^{(k+1)}$. Simultaneously, we use the interval arithmetic to obtain the other box about the propagated parallelepiped image, $IA(\mathbb{B}_\infty^{(k)})$ as in [13]:

$$IA(\mathbb{B}_\infty^{(k)}) = \text{midrad}(\tilde{\boldsymbol{m}}^{(k)}, \tilde{\boldsymbol{r}}^{(k)}) \text{ s.t. } \tilde{\boldsymbol{m}}^{(k)} = h^{(k+1)}(\boldsymbol{m}^{(k)}), \ \tilde{\boldsymbol{r}}^{(k)} = |\mathbf{W}^{(k+1)}| \, \boldsymbol{r}^{(k)}, \quad (7)$$

where $|\mathbf{W}|$ takes the element-wise absolute values of $\mathbf{W}$. The above two propagations can be easily extended to nonlinear layers. The details are described in the supplementary material.

Finally, we can obtain the box outer bound $\mathbb{B}_\infty^{(k+1)} = out_\infty(h^{(k+1)}(\mathbb{B}_2^{(k)})) \cap IA(\mathbb{B}_\infty^{(k)})$ for the next $(k+1)$-th layer as illustrated in Figure 1 with the following equations:

$$\boldsymbol{m}^{(k+1)} = (\boldsymbol{ub}^{(k+1)} + \boldsymbol{lb}^{(k+1)})/2, \ \boldsymbol{r}^{(k+1)} = (\boldsymbol{ub}^{(k+1)} - \boldsymbol{lb}^{(k+1)})/2 \text{ s.t.}$$
$$\boldsymbol{ub}^{(k+1)} = \max(\hat{\boldsymbol{m}}^{(k)} + \hat{\boldsymbol{r}}^{(k)}, \tilde{\boldsymbol{m}}^{(k)} + \tilde{\boldsymbol{r}}^{(k)}), \ \boldsymbol{lb}^{(k+1)} = \min(\hat{\boldsymbol{m}}^{(k)} - \hat{\boldsymbol{r}}^{(k)}, \tilde{\boldsymbol{m}}^{(k)} - \tilde{\boldsymbol{r}}^{(k)}), \quad (8)$$

where $\max$ and $\min$ take the element-wise maximum and minimum values, respectively.

In the penultimate layer, we obtain the intersection $\mathbb{B}_\infty^{(K-1)} \cap \mathbb{B}_2^{(K-1)}$ of the box outer bound $\mathbb{B}_\infty^{(K-1)}$ and the ball outer bound $\mathbb{B}_2^{(K-1)}$. The intersection is propagated to the logit space through the last linear layer to obtain the tight outer bound, as $\hat{z}(\mathbb{B}(\boldsymbol{x})) = h^{(K)}(\mathbb{B}_\infty^{(K-1)} \cap \mathbb{B}_2^{(K-1)}) \subset \mathcal{Z}$.

**Extension to $\ell_p$-norm**  We note that BCP can be easily extended to $\ell_p$-certifiable training for any $p > 0$ by modifying $\mathbb{B}_2^{(0)} = \mathbb{B}_2(\boldsymbol{x}, \epsilon)$ to $\mathbb{B}_2(\boldsymbol{x}, \epsilon')$ circumscribing $\mathbb{B}_p(\boldsymbol{x}, \epsilon)$ in the input space $\mathbb{R}^N$, where $\epsilon' = N^{1/2-1/max(p,2)}\epsilon$. For the $\ell_\infty$-case, we can use $\mathbb{B}_2^{(0)} = \mathbb{B}_2(\boldsymbol{x}, \sqrt{N}\epsilon)$ circumscribing $\mathbb{B}_\infty^{(0)}$. Thus, for $\ell_\infty$-bound, BCP can be considered as a generalized version of IBP (Interval Bound Propagation) [13]. We found that BCP shows a similar performance to IBP as an $\ell_\infty$-certified training (see the supplementary material for the details). For now we focus on the performance of BCP under $\ell_2$-perturbations.

## 3.2 Certifiable Training Algorithm

**Formulation**  Our certifiable algorithm aims to minimize the objective $\mathcal{J}(f, \mathcal{D})$ in (4) to get a robust classifier. The objective contains the worst-translated logit $\underline{z}(\boldsymbol{x})$, which requires computation of the translation vector $t(\boldsymbol{x})$ in (2). In this section, we propose an efficient algorithm to calculate $t(\boldsymbol{x})$ for the tight propagated outer bound $\hat{z}(\mathbb{B}(\boldsymbol{x})) = h^{(K)}(\mathbb{B}_\infty^{(K-1)} \cap \mathbb{B}_2^{(K-1)})$ proposed in Section 3.1. In Equation (2), $z_y(\boldsymbol{x}) - z_m(\boldsymbol{x})$ is easily obtained by a forward pass through the network. However, the optimization $\min_{\boldsymbol{\zeta} \in \hat{z}(\mathbb{B}(\boldsymbol{x}))} (\zeta_y - \zeta_m)$ is nontrivial and dependent on the outer bound $\hat{z}(\mathbb{B}(\boldsymbol{x}))$.

Without loss of generality, we can assume that $y = 1$ and $m = 0$. Then, the optimal values $\zeta_0^*, \zeta_1^*$ are as follows:

$$\zeta_0^*, \zeta_1^* = \underset{(\zeta_0, \zeta_1) \in \Pi_{0,1}\hat{z}(\mathbb{B}(\boldsymbol{x}))}{\text{argmin}} (\zeta_1 - \zeta_0), \quad (9)$$

where $\Pi_{0,1}$ is the projection onto the $\zeta_0\zeta_1$-plane. Then, we can formulate the following optimization:

$$\min_{\boldsymbol{\zeta} \in \hat{z}(\mathbb{B}(\boldsymbol{x}))} (\boldsymbol{e}_1 - \boldsymbol{e}_0)^T \boldsymbol{\zeta} = \min_{\boldsymbol{\zeta}' \in \mathbb{B}_2^{(K-1)} \cap \mathbb{B}_\infty^{(K-1)}} (\boldsymbol{e}_1 - \boldsymbol{e}_0)^T h^{(K)}(\boldsymbol{\zeta}')$$
$$= \min_{\boldsymbol{\zeta}' \in \mathbb{B}_2^{(K-1)} \cap \mathbb{B}_\infty^{(K-1)}} (\mathbf{W}_{1,:}^{(K)} - \mathbf{W}_{0,:}^{(K)})\boldsymbol{\zeta}' + b_1^{(K)} - b_0^{(K)}, \quad (10)$$

where $\boldsymbol{e}_i$ is the $i$-th standard basis vector, and $\mathbf{W}^{(K)}$ and $\boldsymbol{b}^{(K)}$ is the weight matrix and the bias vector for the last linear layer $h^{(K)}$. Note that $\boldsymbol{\zeta}'$ is the vector in the penultimate layer. Therefore, we can construct the following optimization problem that finds the largest violation of the specification to verify the network:

$$\min_{\boldsymbol{\zeta}'} \ \mathbf{c}^T \boldsymbol{\zeta}' \text{ s.t. } \|\boldsymbol{\zeta}' - \mathbf{z}^{(K-1)}\|_2 \le \rho^{(K-1)}, \ |\boldsymbol{\zeta}' - \boldsymbol{m}^{(K-1)}| \le \boldsymbol{r}^{(K-1)}, \quad (11)$$

where $\mathbf{c}$ is a specification vector with $\mathbf{c}^T = \mathbf{W}_{1,:}^{(K)} - \mathbf{W}_{0,:}^{(K)}$, and the second constraint takes the element-wise absolute value and the element-wise inequality. Since it is computationally expensive to

---

**Algorithm 1** Box Constraint Propagation (BCP) Certifiable Training

---

**Input:** training data $(\boldsymbol{x}, y) \sim \mathcal{D}$, target perturbation size $\epsilon_{target}$, network parameterized by $\theta$
**Output:** Robust network $f_\theta$
**repeat**
    Read mini-batch $B$ from $\mathcal{D}$ and adjust $\epsilon$ and $\lambda$ according to the schedule.
    *// Compute the box outer bound and the ball outer bound //*
    $\mathbb{B}_\infty^{(K-1)} = \text{midrad}(\boldsymbol{m}^{(K-1)}, \boldsymbol{r}^{(K-1)})$ where $\boldsymbol{m}^{(K-1)}, \boldsymbol{r}^{(K-1)} = \text{BCP}(\boldsymbol{x}, \epsilon; \theta)$ ((6)-(8)).
    $\mathbb{B}_2^{(K-1)} = \mathbb{B}_2(\mathbf{z}^{(K-1)}, \rho^{(K-1)})$ where $\mathbf{z}^{(K-1)} = h^{(K-1)} \circ \cdots \circ h^{(1)}(\boldsymbol{x})$ and $\rho^{(K-1)} = \epsilon \prod_{i=1}^{K-1} L^{(i)}$
    *// Solve the optimization in* (11) *for each* $m \neq y$ *in parallel //*
    Initialize $\boldsymbol{p} = \mathbf{z}^{(K-1)} - \rho^{(K-1)} \frac{\mathbf{c}}{\|\mathbf{c}\|}$.
    **while not** $|\boldsymbol{p} - \boldsymbol{m}^{(K-1)}| \leq \boldsymbol{r}^{(K-1)}$ **do**
        Decompose $\boldsymbol{p}$ into two parts: $\boldsymbol{p} = \boldsymbol{p}[I] + \boldsymbol{p}[I^c]$, where $I \equiv \{l : |p_l - m_l^{(K-1)}| \geq r_l^{(K-1)}\}$.
        **First phase** Project $\boldsymbol{p}[I]$ onto $\mathbb{B}_\infty^{(K-1)}$.
        **Second phase** With the scaling parameter $\eta$ in (12), update $\boldsymbol{p} \leftarrow \Pi_{\mathbb{B}^{(K-1)}} \boldsymbol{p}[I] + \eta \boldsymbol{p}[I^c]$.
    **end while**
    Calculate the worst-translated logit $\underline{z}(\boldsymbol{x}) = z(\boldsymbol{x}) + t(\boldsymbol{x})$ with (2) and (10):
        $t_m(\boldsymbol{x}) = \mathbf{c}^T(\mathbf{z}^{(K-1)} - \boldsymbol{p})$.
    *// Update Parameters //*
    Update the parameter $\theta$ with the objective (13):
        $\theta \leftarrow \theta - \alpha \nabla_\theta \mathcal{J}(f_\theta, B; \lambda)$.
**until** training phase ends

---

integrate a typical optimization tool within the training loop, we propose a simple iterative algorithm that approaches to the optimal solution of (11) in a finite number of steps. We emphasize that by solving (11) we can obtain a better certificate than the global Lipschitz-based certificate because it uses additional box constraint, $|\boldsymbol{\zeta}' - \boldsymbol{m}^{(K-1)}| \leq \boldsymbol{r}^{(K-1)}$. We will see how this additional constraint affects the outer bound and the verification performance in Section 4.

**Solving the optimization**    We solve the optimization (11) by using an efficient iterative algorithm that terminates when none of the elements violate the box constraint. Our algorithm starts with the initial point $\boldsymbol{p} = \mathbf{z}^{(K-1)} - \rho^{(K-1)} \frac{\mathbf{c}}{\|\mathbf{c}\|}$ which is the optimal solution of (11) when ignoring the box constraint. Then $\boldsymbol{p}$ satisfies the ball constraint but is not guaranteed to satisfy the box constraint. We decompose the indices of $\boldsymbol{p}$ into two parts, $I$ and $I^c$, where $I \equiv \{l : |p_l - m_l^{(K-1)}| \geq r_l^{(K-1)}\}$. Then, we can represent $\boldsymbol{p} = \boldsymbol{p}[I] + \boldsymbol{p}[I^c]$, where $\boldsymbol{p}[J] = \sum_{l \in J} p_l \boldsymbol{e}_l$. Note that $I$ or $I^c$ can be empty, and we define $\boldsymbol{p}[\phi] = \mathbf{0}$. Then, we iterate the following two phases to find the optimal solution efficiently. In the first phase, $\boldsymbol{p}[I]$ is projected onto the box, denoted by $\Pi_{\mathbb{B}^{(K-1)}} \boldsymbol{p}[I]$. In the second phase, $\boldsymbol{p}[I^c]$ is scaled with an adaptive parameter $\eta \geq 1$, as computed by:

$$\eta = \frac{\sqrt{\left(\rho^{(K-1)}\right)^2 - \|\Pi_{\mathbb{B}^{(K-1)}} \boldsymbol{p}[I] - \mathbf{z}^{(K-1)}[I]\|^2}}{\|\boldsymbol{p}[I^c] - \mathbf{z}^{(K-1)}[I^c]\|}. \tag{12}$$

Based on (12), the next point $\boldsymbol{p} \leftarrow \Pi_{\mathbb{B}^{(K-1)}} \boldsymbol{p}[I] + \eta \boldsymbol{p}[I^c]$ is on the boundary $\partial \mathbb{B}_2^{(K-1)}$ of the ball $\mathbb{B}_2^{(K-1)}$ when $I^c \neq \phi$. We skip the scaling in the case of $I^c = \phi$. This iterative algorithm terminates when $\boldsymbol{p}$ satisfies the box constraint. The following proposition shows that our algorithm terminates within a finite step which is determined by the number of elements in $\mathbf{c}$.

**Proposition 2.** *The while loop in Algorithm 1 finds the optimal solution $\boldsymbol{p} = (\boldsymbol{\zeta}')^*$ of the optimization problem* (11) *in a finite number of iterative steps less than the number of elements in* $\mathbf{c}$.

*Proof.* The proof is deferred to the supplementary material. □

Algorithm 1 illustrates the BCP training algorithm. Similar to Kurakin et al. [19], we train on a mixture of normal logit $z(\boldsymbol{x})$ and the worst logit $\underline{z}(\boldsymbol{x})$ as follows:

$$\mathcal{J}(f, \mathcal{D}; \lambda) = \mathbb{E}_{(\boldsymbol{x}, y) \sim \mathcal{D}} \left[ (1 - \lambda) \mathcal{L}(z(\boldsymbol{x}), y) + \lambda \mathcal{L}(\underline{z}(\boldsymbol{x}), y) \right]. \tag{13}$$

We gradually increase the perturbation $\epsilon$ from 0 to the target bound $\epsilon_{target}$ and increase $\lambda$ in (13) from 0 to 1, stabilizing the initial phase of training and improving natural accuracy [13, 44]. Therefore, our algorithm enables fast certifiable training of the robust model with a tight outer bound and is, thus, scalable to large networks.

## 4  Experiments

We demonstrate that the proposed method can provide a tight outer bound for $\ell_2$-perturbation set and train certifiably robust networks, comparing its performance against state-of-the-art certifiable training methods (LMT [36], CAP [40], and IBP [13]) on MNIST and CIFAR10. Moreover, we also show that the BCP scheme can scale to Tiny ImageNet and obtain a meaningful verification accuracy.[1] We further investigate the robustness under a wide range of perturbation. The details of hyper-parameters and architectures used in the experiments can be found in the supplementary material.

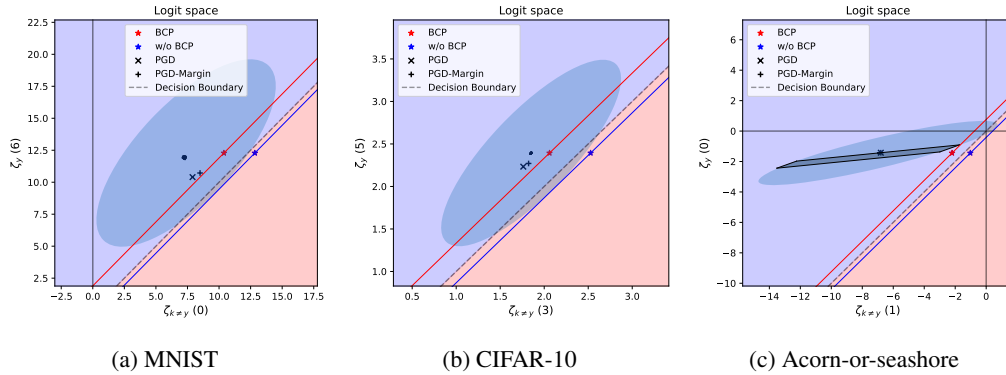

(a) MNIST        (b) CIFAR-10        (c) Acorn-or-seashore

Figure 2: Illustration of the outer bounds for the BCP trained models on (a) MNIST, (b) CIFAR-10, and (c) Acorn-or-seashore classification tasks. BCP cuts off the lower area under the red line from the elliptic area and tightens the outer bound. The shaded parallelogram area in (c) indicates the image of the feasible region for the box constraint after the last linear layer.

**Visualization of Tightening Effects**  Figure 2 illustrates how BCP can tighten the outer region by introducing the box constraint $\mathbb{B}_\infty^{(K-1)}$ in (11). We can easily visualize the high-dimensional ellipsoid $h^{(K)}(\mathbb{B}_2^{(K-1)}) \subset \mathbb{R}^c$ in 2D plane with $\zeta_y$- and $\zeta_{m'}$-axes by projection, where $m'$ corresponds with the most probable class except the true class $y$. However, a high-dimensional parallelogram $h^{(K)}(\mathbb{B}_\infty^{(K-1)})$ is hard to visualize in the 2D plane. Thus, we use the red lines in Figure 2 (a)-(b) to indicate that the projection of the outer region $h^{(K)}(\mathbb{B}_2^{(K-1)} \cap \mathbb{B}_\infty^{(K-1)})$ must lie above the red line and inside the ellipsoid, showing how much area is cut off by the box constraint $\mathbb{B}_\infty^{(K-1)}$. We compute the worst-case margin ($\zeta_y - \zeta_{m'} \geq \zeta_y^* - \zeta_{m'}^*$) based on (9) and build the verification boundary with it, where the red line is obtained from the solution $\zeta_y^*, \zeta_{m'}^*$ of (9) for $\hat{z}(\mathbb{B}(\boldsymbol{x})) = h^{(K)}(\mathbb{B}_2^{(K-1)} \cap \mathbb{B}_\infty^{(K-1)})$, and the blue line is for $\hat{z}(\mathbb{B}(\boldsymbol{x})) = h^{(K)}(\mathbb{B}_2^{(K-1)})$. To verify the network, we utilize the verification boundary, where the verification for $\mathbb{B}_2(\boldsymbol{x}, \epsilon_{target})$ succeeds if the verification boundary is above the decision boundary ($\zeta_y = \zeta_{m'}$). Figure 2c explicitly illustrates the ellipsoid $h^{(K)}(\mathbb{B}_2^{(K-1)})$ and the parallelogram $h^{(K)}(\mathbb{B}_\infty^{(K-1)})$ with the verification boundary for a toy binary classification problem between 'acorn' and 'seashore' derived from Tiny ImageNet dataset. We also indicate the logits for the adversarial examples against PGD attacks based on cross-entropy loss (PGD) and margin-based loss (PGD-Margin), which cannot go over the verification boundaries.

**Quantitative Analysis of Tightness of Outer Bounds**  To quantitatively analyze how much BCP can tighten the outer bound, we use "normalized $\ell_1$-norm" of the translation vector as a measure of

tightness of the outer bound $\hat{z}(\mathbb{B}(\boldsymbol{x}))$ for given input $\boldsymbol{x}$, defined as $\tau(\boldsymbol{x}) = \sum_{i \neq j} t_i(\boldsymbol{x}; j)/c(c-1)$. Without BCP, this is a constant, $\tau = \sum_{i \neq j} \rho^{(K-1)} \|\mathbf{W}_{i,:}^{(K)} - \mathbf{W}_{j,:}^{(K)}\|_2/c(c-1)$, over $\mathcal{X}$ since it only considers the global Lipschitz constant and does not depend on the input. We indicate this constant tightness measure $\tau$ for each dataset as the dotted lines in Figure 3. On the other hand, using BCP, we can consider the local properties of inputs, and thus, we can obtain different tightness for each input. As shown in the violin plots of the tightness in Figure 3, BCP can tighten the outer bounds by 55.4% (MNIST), 45.8% (CIFAR-10), and 25.3% (Tiny ImageNet) on average.

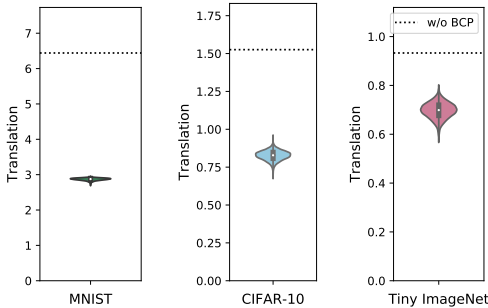

Figure 3: Violin plots of the tightness of the outer bounds. The dotted lines indicate the tightness without BCP. A smaller value indicates a better tightness.

Table 1: Computation time compared to CAP [40]. BCP is over 12 times faster than CAP (* For WRN, CAP uses two GPUs because of the memory limit).

| Data | Structure | Computation time (s/epoch) | | Speed up |
|---|---|---|---|---|
| | | CAP | BCP | |
| MNIST | 4C3F | 689 | **57.5** | ×12.0 |
| CIFAR-10 | 4C3F | 645 | **53.0** | ×12.2 |
| | 6C2F | 1,369 | **56.5** | ×24.2 |
| | WRN | 1,121* | **89.5** | ×12.5 |
| Tiny ImageNet | 8C2F | - | **3,268** | - |

**Verification performance** We evaluate our certifiable training algorithm and other state-of-the-art methods (LMT [36], CAP [40], and IBP [13]) with $\epsilon_{target} = 1.58, 36/255$, and $36/255$ on MNIST, CIFAR-10 and Tiny ImageNet, respectively. We use the same bound for evaluation, $\epsilon_{eval} = \epsilon_{target}$. For MNIST, BCP outperforms other methods not only in terms of verification accuracy but also in terms of standard accuracy. For CIFAR-10, BCP outperforms LMT and IBP, and produces comparable performance with CAP in terms of verification accuracy, whereas outperforming in terms of both standard accuracy and robust accuracy against PGD. For Tiny ImageNet, BCP can achieve a verification accuracy of 20.08%, while LMT and IBP learn constant models and CAP is not scalable to Tiny ImageNet.

To further investigate robustness of the models, in Figure 4, we demonstrate the change of verification accuracy for different $\ell_2$-perturbations $\epsilon_{eval}$. We train the robust models with $\epsilon_{target} = 1.58$ on MNIST (Figure 4a) and $\epsilon_{target} = 36/255$ and $2\epsilon_{target} = 72/255$ on CIFAR-10 (Figure 4b,4c). Comparing to state-of-the-art methods, BCP achieves the highest verification accuracy in a wide

Table 2: Comparison to other verifiable training methods. Best performances are highlighted in bold.

| Data | Structure | # parameters | Method | Accuracy (%) | | |
|---|---|---|---|---|---|---|
| | | | | Standard | PGD | Verification |
| MNIST | 4C3F | 1974762 | CAP | 88.39 | 62.25 | 43.95 |
| | | | LMT | 86.48 | 53.56 | 40.55 |
| | | | BCP | **92.41** | **64.70** | **47.95** |
| CIFAR-10 | 4C3F | 2466858 | CAP | 60.14 | 55.67 | **50.29** |
| | | | LMT | 56.49 | 49.83 | 37.20 |
| | | | IBP | 34.50 | 31.79 | 24.39 |
| | | | BCP | **63.88** | **58.75** | 49.58 |
| | 6C2F | 2250378 | CAP | 60.10 | 56.20 | 50.87 |
| | | | LMT | 63.05 | 58.32 | 38.11 |
| | | | IBP | 32.96 | 31.08 | 23.42 |
| | | | BCP | **65.72** | **60.78** | **51.30** |
| | WRN | 4214850 | CAP | 60.70 | 56.77 | **51.63** |
| | | | LMT | 61.33 | 56.39 | 33.35 |
| | | | BCP | **64.79** | **59.16** | 50.33 |
| Tiny ImageNet | 8C2F | 4342984 | BCP | **28.76** | **26.64** | **20.08** |

range of $\epsilon_{eval}$. The verification accuracy of BCP slowly decreases as increasing $\epsilon$, and the decrease seems almost linear, while we observe a significant drop in verification accuracy when $\epsilon_{eval} \geq \epsilon_{target}$ for CAP. We emphasize that the verification accuracy against a range of perturbation involves more meaningful understanding of robustness than the verification performance at a specific perturbation bound $\epsilon_{eval}$ in Table 2 (see the supplementary material for more detailed results).

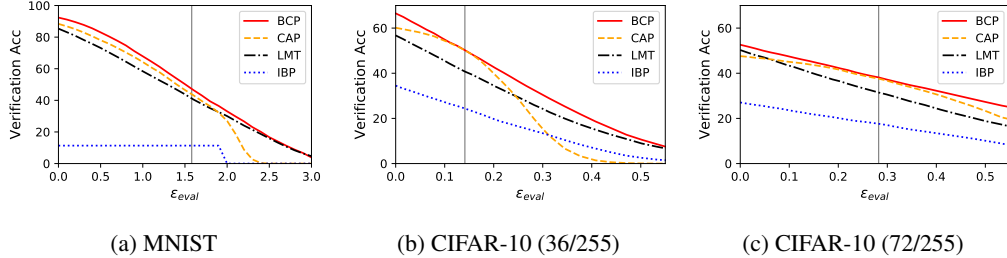

(a) MNIST        (b) CIFAR-10 (36/255)        (c) CIFAR-10 (72/255)

Figure 4: Verification performances of verifiable training methods. The vertical lines indicate $\epsilon_{target}$.

**Computational Cost**    Table 1 shows that BCP is over 12 times faster than CAP. We evaluate the computation times on a single Titan X GPU. For a fair comparison, we use the same batch size for both methods as 50 on MNIST and CIFAR-10 and 5 on Tiny ImageNet. Because CAP is memory-inefficient, they cannot increase the batch size, whereas we can further speed up with a larger batch size. In the case of WRN [42] on CIFAR-10, we can speed up to 61.1 sec/epoch using batch size of 128, while CAP needs two GPUs to run with a batch size of 50. It implies that our certifiable training is efficiently applicable to a large-scale dataset.

## 5   Conclusion

In this study, we propose a fast certifiable training with a tight outer bound. To obtain a tight outer bound, we propose BCP that efficiently computes box constraints which can tighten the outer bound. Then, we train a certifiably robust model by minimizing the certificate loss based on the worst-translated logit over the tight outer bound. By doing so, we can build the first certifiable robust model on Tiny ImageNet under the $\ell_2$-perturbation. We hope that our method can serve as a strong benchmark for certifiable training on a large-scale dataset.

## Broader Impact

Verifiable training can be used as one of a general learning scheme for applications to security-sensitive domains such as self-driving cars, face recognition, and medical diagnostics. In these applications, an adversarial example is a potential safety hazard that we want to avoid. By training a model with BCP, we can guarantee that no adversarial attack within a given norm-based perturbation can break the model. However, we should note that there is a trade-off between security and performance. Our work tends to lean to the security aspect, having relatively low accuracy on natural data. The sacrifice of performance can halve the benefits of applying deep learning models, and security concerns can restrain deployments in the real system. We are already familiar with deep learning models embedded in our everyday products or services, such as a smart speaker, ridesharing apps, and social media services. Therefore, a balance of performance and security is required depending on the characteristics of the application. The development of verifiable training algorithm enables to improve standard accuracy, exactly quantifying the security. In addition, the quantification of performance and security can help to adjust the balance between them.

## Acknowledgement

This research was supported by Basic Science Research Program through the National Research Foundation of Korea (NRF) funded by the Ministry of Education (2018R1D1A1A02085851), and in part by the NRF Grant funded by the Korean Government (MSIT) (NRF-2019R1A2C2002358). The corresponding author is Saerom Park.

## Footnotes

[1]https://tiny-imagenet.herokuapp.com/

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
