[Supplementary Material]

# Supplementary Material for Lipschitz-Certifiable Training with a Tight Outer Bound

**Sungyoon Lee**
Seoul National University
Seoul, Korea
goman1934@snu.ac.kr

**Jaewook Lee**
Seoul National University
Seoul, Korea
jaewook@snu.ac.kr

**Saerom Park**
Sungshin Women's University
Seoul, Korea
psr6275@sungshin.ac.kr

## A  The proofs of the propositions

**Proposition 1.** *For an outer bound $\hat{z}(\mathbb{B}(\boldsymbol{x})) \supset z(\mathbb{B}(\boldsymbol{x}))$ and its corresponding worst-translated logit $\underline{z}(\boldsymbol{x})$, the following inequality holds:*

$$\max_{\boldsymbol{\zeta} \in \hat{z}(\mathbb{B}(\boldsymbol{x}))} \mathcal{L}(\boldsymbol{\zeta}, y) \leq \mathcal{L}(\underline{z}(\boldsymbol{x}), y), \tag{S1}$$

*where $\mathcal{L}$ is the cross-entropy loss function.*

*Proof.*

$$
\begin{aligned}
&\max_{\boldsymbol{\zeta} \in \hat{z}(\mathbb{B}(\boldsymbol{x}))} \mathcal{L}(\boldsymbol{\zeta}, y) \\
&= \log\left(1 + \max_{\boldsymbol{\zeta} \in \hat{z}(\mathbb{B}(\boldsymbol{x})))} \sum_{k \neq y} \exp\left(-(\zeta_y - \zeta_k)\right)\right) \\
&\leq \log\left(1 + \sum_{k \neq y} \max_{\boldsymbol{\zeta} \in \hat{z}(\mathbb{B}(\boldsymbol{x})))} \exp\left(-(\zeta_y - \zeta_k)\right)\right) \\
&= \log\left(1 + \sum_{k \neq y} \exp\left(-\min_{\boldsymbol{\zeta} \in \hat{z}(\mathbb{B}(\boldsymbol{x}))} (\zeta_y - \zeta_k)\right)\right) \\
&= \mathcal{L}(\underline{z}(\boldsymbol{x}), y)
\end{aligned}
$$

$\square$

**Proposition 2.** *The while loop in Algorithm 1 finds the optimal solution $\boldsymbol{p} = (\boldsymbol{\zeta}')^*$ of the optimization problem* (11) *in a finite number of iterative steps less than the number of elements in* **c**.

*Proof.* We denote the number of elements in **c** as $N_c$. For $n$-th iteration, we denote $I = \{l : |p_l - m_l^{(K-1)}| \geq r_l^{(K-1)}\}$ as $I_n$. Then, for each iteration in the while loop, at least one index does not satisfies the box constraints, i.e., $\exists i$ s.t. $|p_i - m_i^{(K-1)}| > r_i^{(K-1)}$, and the index $i$ is added to $I_n$. And once an index $i$ is added to $I_n$ then the $i$-th elements $p_i$ is projected on the box $\mathbb{B}_\infty^{(K-1)}$ and after that it stays in $I_k$ for $k \geq n$, i.e., $I_n$ is a strictly increasing sequence of sets. And if $|I_n| = N_c$, then after the first phase of the iteration, there is no element that violates the box constraints, and the iteration stops (Note that when $|I_n| = N_c$, i.e., $I_n^c = \phi$, we skip the second phase). Therefore, $n \leq N_c$, i.e., our iterative algorithm stops within a finite number of iterative steps less than $N_c$. Now, to prove the proposition, it is enough to show the optimality of the final $\boldsymbol{p}$.

Without loss of generality, we can assume that $\mathbf{z}^{(K-1)} = 0$, $\rho^{(K-1)} = 1$ and $\mathbf{c} \leq 0$. Then we have $\boldsymbol{lb}^{(K-1)} \leq 0 \leq \boldsymbol{ub}^{(K-1)}$, and the final $\boldsymbol{p}$ satisfies $\boldsymbol{p} \geq 0$.

We put $\boldsymbol{v} = -\mathbf{c} \in \mathbb{R}^{N_c}$ and $J \equiv \{l : p_l = ub_l^{(K-1)}\}$. Then the final $\boldsymbol{p} = \boldsymbol{p}[J] + \boldsymbol{p}[J^c]$ satisfies $\boldsymbol{p}[J] \leq \alpha \boldsymbol{v}[J]$ and $\boldsymbol{p}[J^c] = \alpha \boldsymbol{v}[J^c]$ for some $\alpha \geq 1$ that is the product of all previous $\eta$'s. We want to prove $\boldsymbol{p}$ is a local minimum of (11), then since (11) is a convex optimization, we can prove that $\boldsymbol{p}$ is the global optimum. We consider a closed local area $\mathbb{B}(\boldsymbol{p}, \delta > 0)$ such that for any $\boldsymbol{q} \in \mathbb{B}(\boldsymbol{p}, \delta)$, $\boldsymbol{q} \geq 0$ and we can ignore the box constraint for $q_l$ for $l \in J^c$. We call a local optimal solution of (11) in $\mathbb{B}(\boldsymbol{p}, \delta)$ as $\boldsymbol{p}^*$. If $\boldsymbol{p}^*[J] = \boldsymbol{p}[J]$, $\boldsymbol{v}^T(\boldsymbol{p} - \boldsymbol{p}^*) = \boldsymbol{v}^T[J^c](\boldsymbol{p}[J^c] - \boldsymbol{p}^*[J^c]) = \alpha\|\boldsymbol{v}[J^c]\|^2 - \|\boldsymbol{v}[J^c]\|\|\boldsymbol{p}^*[J^c]\|\cos\phi = \|\boldsymbol{v}[J^c]\|(\|\boldsymbol{p}[J^c]\| - \|\boldsymbol{p}^*[J^c]\|\cos\phi) \geq 0$ since $\|\boldsymbol{p}[J^c]\| = \sqrt{1 - \|\boldsymbol{p}[J]\|^2} = \sqrt{1 - \|\boldsymbol{p}^*[J]\|^2} \geq \sqrt{\|\boldsymbol{p}^*\|^2 - \|\boldsymbol{p}^*[J]\|^2} = \|\boldsymbol{p}^*[J^c]\|$ where $\phi$ is the angle between the two vectors, $\boldsymbol{p}^*[J^c]$ and $\boldsymbol{v}[J^c]$. Thus $\boldsymbol{p}$ is a local optimal.

Therefore, to prove the proposition with a proof by contradiction, we suppose $\boldsymbol{p}^*[J] \neq \boldsymbol{p}[J]$, i.e., there is an index $j$ such that $p_j^* < ub_j^{(K-1)}$. If $J^c = \phi$, then it contradicts the optimality of $\boldsymbol{p}^*$ since $\boldsymbol{v}^T\boldsymbol{p} > \boldsymbol{v}^T\boldsymbol{p}^*$. Therefore, we can further assume $J^c \neq \phi$, and thus $\|\boldsymbol{p}\| = 1$. Moreover, if $\|\boldsymbol{p}^*\| < 1$, then we can further extend $\boldsymbol{p}^*[J^c]$ to produce a larger inner product with $\boldsymbol{v}$, and this contradicts the assumption. Thus, $\|\boldsymbol{p}\| = \|\boldsymbol{p}^*\| = 1$ and $\|\boldsymbol{p}[J^c]\| < \|\boldsymbol{p}^*[J^c]\|$. We say $(p_j^*)^2 + \|\boldsymbol{p}^*[J^c]\|^2 = r^2$. Then we consider a two-dimensional space $\mathbf{U}$ spanned by two orthonormal vectors, $\boldsymbol{e}_j$ and $\boldsymbol{p}^*[J^c]/\|\boldsymbol{p}^*[J^c]\|$, say $\boldsymbol{u}^{(1)}$ and $\boldsymbol{u}^{(2)}$. Then $\boldsymbol{v}^{(1)} = \Pi_{\mathbf{U}}\boldsymbol{p}^* = p_j^*\boldsymbol{u}^{(1)} + \|\boldsymbol{p}^*[J^c]\|\boldsymbol{u}^{(2)} = r\cos\theta_1\boldsymbol{u}^{(1)} + r\sin\theta_1\boldsymbol{u}^{(2)}$ is on the sphere that has radius $r > 0$. We consider another vector on the sphere, $\boldsymbol{v}^{(2)} = ub_j^{(K-1)}\boldsymbol{u}^{(1)} + \beta\boldsymbol{u}^{(2)} = r\cos\theta_2\boldsymbol{u}^{(1)} + r\sin\theta_2\boldsymbol{u}^{(2)}$ with $\beta \geq 0$. Then, $\alpha\boldsymbol{v}$ projected on $\mathbf{U}$ is $\Pi_{\mathbf{U}}(\alpha\boldsymbol{v}) = (\alpha\boldsymbol{v}^T\boldsymbol{u}^{(1)})\boldsymbol{u}^{(1)} + (\alpha\boldsymbol{v}^T\boldsymbol{u}^{(2)})\boldsymbol{u}^{(2)} = (\alpha\boldsymbol{v}^T\boldsymbol{u}^{(1)})\boldsymbol{u}^{(1)} + (\boldsymbol{p}^T\boldsymbol{u}^{(2)})\boldsymbol{u}^{(2)}$ with $\alpha\boldsymbol{v}^T\boldsymbol{u}^{(1)} \geq ub_j^{(K-1)}$ because $j \in J$ and $ub_j^{(K-1)} = p_j \leq \alpha v_j$. And, $\boldsymbol{p}$ projected on $\mathbf{U}$ is $\Pi_{\mathbf{U}}\boldsymbol{p} = (\boldsymbol{p}^T\boldsymbol{u}^{(1)})\boldsymbol{u}^{(1)} + (\boldsymbol{p}^T\boldsymbol{u}^{(2)})\boldsymbol{u}^{(2)}$ with $\boldsymbol{p}^T\boldsymbol{u}^{(2)} \leq {\boldsymbol{v}^{(2)}}^T\boldsymbol{u}^{(2)} = \beta$ since $\|\boldsymbol{v}^{(2)}\| = \|\Pi_{\mathbf{U}}\boldsymbol{p}^*\| \geq \|\Pi_{\mathbf{U}}\boldsymbol{p}\|$. We can write $\boldsymbol{v}^T(a\boldsymbol{u}^{(1)} + b\boldsymbol{u}^{(2)}) = av_j + b\boldsymbol{v}[J^c]^T\boldsymbol{u}^{(2)} = ar_1\cos\theta_0 + br_1\cos\theta_0$ with $r_1 > 0$. Thus $\boldsymbol{v}^T\boldsymbol{v}^{(i)} = rr_1(\cos\theta_0\cos\theta_i + \sin\theta_0\sin\theta_i) = rr_1\cos(\theta_i - \theta_0)$ for $i = 1, 2$. Since $0 \leq p_j^* < ub_j^{(K-1)} \leq \alpha\boldsymbol{v}^T\boldsymbol{u}^{(1)}; 0 \leq \alpha\boldsymbol{v}^T\boldsymbol{u}^{(2)} = \boldsymbol{p}^T\boldsymbol{u}^{(2)} \leq \beta \leq \|\boldsymbol{p}^*[J^c]\|$ and $0 \leq \theta_0 \leq \theta_i \leq \pi/2$, we have $\tan\theta_0 = \alpha\boldsymbol{v}^T\boldsymbol{u}^{(2)}/\alpha\boldsymbol{v}^T\boldsymbol{u}^{(1)} \leq \boldsymbol{p}^T\boldsymbol{u}^{(2)}/ub_j^{(K-1)} \leq \tan\theta_2 = \beta/ub_j^{(K-1)} < \tan\theta_1 = \|\boldsymbol{p}^*[J^c]\|/p_j^*$. Therefore, we have $0 \leq \theta_2 - \theta_0 < \theta_1 - \theta_0 \leq \pi/2$ and $0 \leq \cos(\theta_0 - \theta_1) < \cos(\theta_0 - \theta_2)$. Therefore, $\boldsymbol{v}^T\boldsymbol{v}^{(1)} < \boldsymbol{v}^T\boldsymbol{v}^{(2)}$. However, $\Pi_{\mathbf{U}}\boldsymbol{p}$ is closer to $\boldsymbol{v}^{(2)}$ than to $\boldsymbol{v}^{(1)} = \Pi_{\mathbf{U}}\boldsymbol{p}^*$. Thus, we found $\boldsymbol{p}^*[J - \{j\}] + \boldsymbol{v}^{(2)} \in \mathbb{B}(\boldsymbol{p}, \delta)$ which yield a larger inner product with $\boldsymbol{v}$ than $\boldsymbol{p}^* = \boldsymbol{p}^*[J - \{j\}] + \boldsymbol{v}^{(1)}$ which contradicts the local optimality of $\boldsymbol{p}^*$.

$\square$

## B   Outer Bound Propagation

In this section, we present intuition behind the design of BCP and the deferred explanation for calculating outer bound propagation such as layer-wise Lipschitz constant, propagated circumscribed box, and extension to a residual layer. We further provide complexity analysis on BCP.

### B.1   Intuition behind BCP

BCP provides a tight outer bound that addresses the overestimation problems of the global Lipschitz constant. An outer bound computed by the global Lipschitz constant is highly overestimated because of the following reasons. First, it is overestimated when propagating through a linear layer. For a linear layer (including convolutional, average pooling, and normalization layers), the layer-wise Lipschitz constant is the maximum eigenvalue of the weight matrix (the factor by which the corresponding eigenvector is scaled); thus, it overestimates stretching along directions except for the corresponding eigenvector. On the other hand, with BCP, the box constraint bound $out_\infty(h^{(k+1)}(\mathbb{B}_2^{(k)}))$ calculates the radius vector $\hat{\boldsymbol{r}}^{(k+1)}$ by considering scaling along all basis axes. Second, the outer bound is overestimated because of ReLU layers. After propagating a ball $\mathbb{B}_2(\boldsymbol{\mu}, \rho)$ through a ReLU layer, we can estimate the propagated outer bound with a new ball $\mathbb{B}_2(\boldsymbol{\mu}^+, \rho)$ where $\boldsymbol{\mu}^+ = \max(\boldsymbol{\mu}, 0)$. However, the true image $\text{ReLU}(\mathbb{B}_2(\boldsymbol{\mu}, \rho))$ has no negative elements. In the case of BCP, the box constraint $IA(\mathbb{B}_\infty^{(k)})$ tightens the image of $\mathbb{B}_\infty^{(k)}$ in the ReLU layers by cutting off the negative regions.

## B.2 Power iteration algorithm

In our algorithm, we apply the power iteration to compute the layer-wise Lipschitz constants efficiently. The obtained Lipschitz constants are used to compute the layer-wise outer bounds, $(\mathbb{B}_2^{(k)}, \mathbb{B}_\infty^{(k)})$. We run 1 iteration per batch during training as mentioned in [3], which is enough since SGD only makes small updates to $\mathbf{W}$ and its spectral norm. On the other hand, we run the power iteration until convergence for inference. Note that once we get the layer-wise Lipschitz constants of the trained model, we don't have to run the iteration again.

In Algorithm S1, we provide the well-known power iteration method which calculates Lipschitz constant for a linear layer to make the main paper self-contained.

---

**Algorithm S1** Power iteration.

---

**Input:** weight $\mathbf{W}$, initial value $\boldsymbol{u}$, maximum iteration $n$
**Output:** spectral norm $\sigma$
Initialize u with a random vector with the same shape of the input for the linear layer if $u$ is not given.
$i \leftarrow 0$
**repeat**
  $\boldsymbol{v} \leftarrow \mathbf{W}\boldsymbol{u}/\|\mathbf{W}\boldsymbol{u}\|_2$
  $\boldsymbol{u} \leftarrow \mathbf{W}^T\boldsymbol{v}/\|\mathbf{W}^T\boldsymbol{v}\|_2$
  $i \leftarrow i+1$
**until** it converges or $i \geq n$
$\sigma \leftarrow \boldsymbol{v}^T\mathbf{W}\boldsymbol{u}$

---

In case of a convolutional layer, we used the revised version of the power iteration method in [3]. Algorithm S2 illustrates the power iteration algorithm for a convolutional layer. We denote the conv_transpose operation as $conv^T$.

---

**Algorithm S2** Convolutional Power iteration [3].

---

**Input:** Convolutional weight $\mathbf{W}$, initial value $\boldsymbol{u}$, maximum iteration $n$
**Output:** spectral norm $\sigma$
Initialize u with a random vector with the same shape of the input for the convolutional layer if $u$ is not given.
$i \leftarrow 0$
**repeat**
  $\boldsymbol{v} \leftarrow conv(\mathbf{W}, \boldsymbol{u})/\|conv(\mathbf{W}, \boldsymbol{u})\|_2$
  $\boldsymbol{u} \leftarrow conv^T(\mathbf{W}, \boldsymbol{v})/\|conv^T(\mathbf{W}, \boldsymbol{v})\|_2$
  $i \leftarrow i+1$
**until** it converges or $i \geq n$
$\sigma \leftarrow \boldsymbol{v} \cdot conv(\mathbf{W}, \boldsymbol{u})$

---

## B.3 The circumscribed box $out_\infty(h^{(k+1)}(\mathbb{B}_2^{(k)}))$

In Section 3.1, we explained the computation of the circumscribed box $out_\infty(h^{(k+1)}(\mathbb{B}_2^{(k)}))$ for the affine transformation $h^{(k+1)}$. In this section, we extend it to nonlinear case with a general proof. For ease of explanation, we re-write the equation (6) for the affine case as follows:

$$out_\infty(h^{(k+1)}(\mathbb{B}_2^{(k)})) = \text{midrad}(\hat{\boldsymbol{m}}^{(k)}, \hat{\boldsymbol{r}}^{(k)}) \text{ s.t.}$$
$$\hat{\boldsymbol{m}}^{(k)} = h^{(k+1)}(\boldsymbol{\mu}^{(k)}), \ \hat{r}_i^{(k)} = \|\mathbf{W}_{i,:}^{(k+1)}\|_2 \, \rho^{(k)}. \tag{S2}$$

*Proof.* It is clear that $\hat{\boldsymbol{m}}^{(k)} = h^{(k+1)}(\boldsymbol{\mu}^{(k)})$. The desired radius $\hat{r}_i^{(k)}$ along $i$-th axis is the maximum of the inner product $\boldsymbol{e}_i^T\mathbf{W}^{(k+1)}x = \mathbf{W}_{i,:}^{(k+1)}\boldsymbol{x}$, where $\|\boldsymbol{x}\|_2 \leq \rho^{(k)}$ and $\boldsymbol{W}_{i,:}^{(k+1)}$ is the $i$-th row of the weight matrix $\mathbf{W}^{(k+1)}$ of the linear function $h^{(k+1)}$. Therefore, $\mathbf{W}_{i,:}^{(k+1)}\boldsymbol{x} \leq \|\mathbf{W}_{i,:}^{(k+1)}\|\|\boldsymbol{x}\|$, and we finally get the radius $\hat{r}_i^{(k)} = \|\mathbf{W}_{i,:}^{(k+1)}\| \, \rho^{(k)}$. $\square$

In the case of nonlinear activation function $\sigma$, we can derive the results $\hat{\boldsymbol{m}}^{(k)} = \sigma(\boldsymbol{\mu}^{(k)}), \hat{r}_i = L(\sigma_i)\rho^{(k)}$ with the Lipschitz constant $L(\sigma_i)$ of the nonlinear function $\sigma_i$, because the desired radius $\hat{r}_i$ is the maximum of the inner product $\boldsymbol{e}_i^T \sigma(\boldsymbol{x}) = \sigma_i(\boldsymbol{x})$, where $\|\boldsymbol{x}\|_2 \leq \rho^{(k)}$. In the case of ReLU activation, we have $\hat{r}_i = \rho^{(k)}$ since the Lipschitz constant $L(\sigma_i)$ is 1.

## B.4   BCP through residual layers

Our method can be applied to a wide range of network architectures including residual networks. In this section, we consider an operation through the residual layer as $h^{(k+1)}(\boldsymbol{x}) = f^{(k+1)}(\boldsymbol{x}) + g^{(k+1)}(\boldsymbol{x})$ for some functions $f^{(k+1)}$ and $g^{(k+1)}$. Therefore, we propagate the pair $(\mathbb{B}_2^{(k)}, \mathbb{B}_\infty^{(k)})$ through $f^{(k+1)}$ and $g^{(k+1)}$ independently, and obtain two pairs $(\mathbb{B}_2^{(k+1),1}, \mathbb{B}_\infty^{(k+1),1})$ and $(\mathbb{B}_2^{(k+1),2}, \mathbb{B}_\infty^{(k+1),2})$, respectively. We denote $\mathbb{B}_2^{(k+1),i} = \mathbb{B}_2(\mathbf{z}^{(k+1),i}, \rho^{(k+1),i})$ and $\mathbb{B}_\infty^{(k+1),i} =$ midrad$(\boldsymbol{m}^{(k+1),i}, \boldsymbol{r}^{(k+1),i})$, where $i = 1, 2$. Finally, we can get the pair $(\mathbb{B}_2^{(k+1)}, \mathbb{B}_\infty^{(k+1)})$ for $h^{(k+1)}$ as follows:

$$
\begin{aligned}
\mathbb{B}_2^{(k+1)} &= \mathbb{B}_2(\mathbf{z}^{(k+1)}, \rho^{(k+1)}), \\
\mathbb{B}_\infty^{(k+1)} &= \text{midrad}(\boldsymbol{m}^{(k+1)}, \boldsymbol{r}^{(k+1)}) \text{ s.t} \\
\mathbf{z}^{(k+1)} &= \mathbf{z}^{(k+1),1} + \mathbf{z}^{(k+1),2}, \\
\rho^{(k+1)} &= \rho^{(k+1),1} + \rho^{(k+1),2}, \\
\boldsymbol{m}^{(k+1)} &= \boldsymbol{m}^{(k+1),1} + \boldsymbol{m}^{(k+1),2}, \\
\boldsymbol{r}^{(k+1)} &= \boldsymbol{r}^{(k+1),1} + \boldsymbol{r}^{(k+1),2}.
\end{aligned}
\tag{S3}
$$

## B.5   Complexity Analysis

In this section, we provide the computational complexity analysis on the proposed algorithm. To simplify the discussion, we consider a $K'$-layered feedforward network which has $K'$ linear layers followed by non-linear activations. We further suppose the network has $n$ neurons for all $K' - 1$ layers except for the output layers which has $c \ll n$ neurons. The simple forward propagation costs $O((K' - 1)n^2 + nc) = O(K'n^2)$ for each input. For the proposed method, it takes $O(2(K' - 1)n^2)$ for (6) and (7). In addition, it takes $O(2s(K' - 1)n^2)$ for the power iteration with the iteration $s(= 1)$, and takes $O(2tn)$ for computing $\eta$ in (12) with the iteration $t \ll n$. Lastly, to compute the worst-translated logit, it takes $O(cn)$. Therefore, the total computation of the BCP costs $O((4 + 2s)(K' - 1)n^2 + 2tn + cn) = O(K'n^2)$ which is only $O(1)$ times slower than regular training. In detail, BCP is about 6 times slower than regular training for the iteration $s = 1$, and we empirically found that it is roughly correct (e.g. 8 vs 53 sec/epoch for 4C3F on CIFAR-10).

## C Experimental details

### C.1 Data Description

MNIST [7]: 10 classes, 600K training images, 100K test images.
CIFAR-10 [6]: 10 classes, 500K training images, 100K test images.
Tiny ImageNet [2]: 200 classes, 100K training images, 10K validation images, 10K test images.

### C.2 Hyper-parameters

We list the hyper-parameters used in the proposed certifiable training in Table S1. They are obtained using grid search. We considered the parameters, learning rate $\in [0.0001, 0.0003, 0.001, 0.003, 0.01]$, the length of the warm-up period $\in [1, 3, 5]$, and the length of the ramp-up period $\in [10, 20, 50]$. We also considered the hyperparameters used in CRONW-IBP [11]. In our certifiable training, we used the objective (13). During warm-up period, we conducted the standard training, i.e. $\lambda = 0$ while during ramp-up period, we gradually increased $\lambda$ from 0 to 1. The sensitivity analysis on the schedule of $\lambda$ will be discussed in Section C.4. For CAP [10], we used the pretrained models given by the authors.[1] We used a single iteration for the power iteration as mentioned in Section B.2, and used 10 iterations for the optimization (11) during training on MNIST. On CIFAR-10, we found that it is enough to run a single iteration for the optimization (11) during training. On Tiny ImageNet, we cannot run more than one iteration because of the memory constraint, but we can obtain a verification accuracy of $20.1\%$. In test phase, we run both iterations until convergence. To evaluate the PGD accuracy, we set the step-size to $\epsilon_{eval}/4$ and the number of iterations to 100.

Table S1: Hyper-parameters used in the certifiable training.

| Data | image size | # class | optimizer | optimizer parameters | epoch | learning rate | warm-up/ ramp-up | weight decay $(\times \gamma)$ |
|---|---|---|---|---|---|---|---|---|
| MNIST | (28,28) | 10 | Adam | $\gamma = 0.1$ | 60 | 0.0003 | 1/20 | [21,30,40] |
| CIFAR-10 | (3,32,32) | 10 | Adam | $\gamma = 0.5$ | 100 | 0.001 | 10/121 | every 10 epochs after 131 |
| Tiny ImageNet | (3,64,64) | 200 | SGD | $\gamma = 0.1$ momentum=0.1 weight decay=$2e^{-4}$ | 100 | 0.001 | 2/50 | [50,70,90] |

### C.3 Network architectures

We denote the convolutional layer with the output channel $c$, the kernel $k$, and the stride $s$ as $C(c, k, s)$ (or $C(c, k, s, p)$ if it uses the padding $p \neq 0$) and the linear layer with the output channel $c$ as $F(c)$. We apply ReLU activation after every convolutional layers and linear layers except for the last linear layer. For brevity, we omit the notation for the ReLU activation layers and the flatten layer before the first linear layer. The network 4C3F is the same as that used in Wong et al. [10].

- 4C3F:
  C(32,3,1,1)-C(32,4,2,1)-C(64,3,1,1)-C(64,4,2,1)-F(512)-F(512)-F(10)
- 6C2F:
  C(32,3,1,1)-C(32,3,1,1)-C(32,4,2,1)-C(64,3,1,1)-C(64,3,1,1)-C(64,4,2,1)-F(512)-F(10)
- 8C2F (Tiny ImageNet):
  C(64,3,1,1)-C(64,3,1,1)-C(64,4,2)-C(128,3,1,1)-C(128,3,1,1)-C(128,4,2)-C(256,3,1,1)-C(256,4,2)-F(256)-F(200)

## C.4 Additional Experiments

**Lipschitz constant.** The Lipschitz constant of a neural network is tightly correlated with the excess risk, the difference between the test error and the training error. This implies that a network with a large Lipschitz constant shows poor generalization performance [1]. However, the Lipschitz constant keeps increasing through standard training as shown in Figure S1 (top). On the other hand, the Lipschitz constant keeps decreasing in the BCP training phase as shown in Figure S1 (bottom). The left dotted vertical line indicates the step when warm-up ends, and the other line indicates the step when the ramp-up ends. It demonstrates that our objective (13) encourages the model to be robust and to generalize better than standard training.

On the other hand, strong constraints on the global Lipschitz constant may reduce the expressive capacity and the performance of the network [5]. As shown in Figure 2 and 3, BCP can compute a tighter outer bound, and thus it can relieve the constraints on the Lipschitz constant of the model.

Figure S2 shows the ratio of the Lipschitz constant of the model trained with BCP ($L_{BCP}^{(-1)} = \Pi_{k=1}^{K-1} L_{BCP}^{(k)}$) to its counterpart trained without BCP ($L_0^{(-1)} = \Pi_{k=1}^{K-1} L_0^{(k)}$). Using BCP had 17.2% larger Lipschitz constant, achieving a higher model capacity as shown in the results of the standard accuracy in Table 2.

Therefore, we can conclude that our certifiable training can keep generalization performance without excessive loss of the expressive capacity.

Figure S1: The change of the Lipschitz constants in a log scale during standard training (top) and the BCP training (bottom).

**Verification Performance** As in Figure 4 in Section 4, we conducted the same analysis on different settings as demonstrated in Figure S3. On MNIST, we use $\epsilon'_{target} = 2$ instead of $2\epsilon_{target}$ because $\epsilon_{target} = 1.58$ is already large enough. The verification accuracies of BCP and LMT slowly decrease as increasing $\epsilon_{eval}$ almost linearly. However, CAP fails to achieve robustness under the large

Figure S2: The change of the ratio of the Lipschitz constant between a model trained with BCP and a model trained without BCP during training.

perturbations for $\epsilon_{target}$ and to train the robust model for $\epsilon'_{target}$. For CIFAR-10, we also tested on 6C2F architecture and found that BCP outperforms the other state-of-the-art methods. Unlike in the case of BCP where the verification accuracy decreases slowly as increasing $\epsilon_{eval}$, the verification accuracy of CAP with $\epsilon_{target}$ drastically decreases when $\epsilon_{eval}$ becomes larger than $\epsilon_{target}$.

We also analyze the trade-off between standard accuracy and certifiable robustness. We used different target bounds $\epsilon_{target} = \{30, 32, 34, 36, 38, 40, 42\}/255$ during training to obtain different models while fixing the evaluation bound with $\epsilon_{eval} = 36/255$. Figure S4 shows the trade-off for 7-layer convolutional networks, 4C3F, trained with BCP, without BCP, and with LMT [9] on CIFAR-10. The true robust classification error is lower bounded by the classification error against PGD [8] and is upper bounded by the verification error. In Figure S4, the $x$-axis indicates standard accuracy $(a_s)$ and on the $y$-axis, we plot line segments representing intervals $[l, u]$ containing the true robust classification accuracy with the lower bound of verification accuracy $(l = a_v)$ and with the upper bound of classification accuracy against PGD $(u = a_{PGD})$. We connected the points, $(a_s, a_v)$, with the dotted lines in Figure S4, demonstrating that BCP has the best trade-off among those methods. The smaller the target bound, the better the standard accuracy achieved. We also found that the robust accuracy against PGD increases with standard accuracy.

**Comparison to IBP [4] and CAP [10] as an $\ell_\infty$-certifiable training**    As mentioned in Section 3.1, BCP can be extended to other norm-bounded perturbations, e.g., $\ell_\infty$-norm. We compare the performance of BCP to IBP and CAP as an $\ell_\infty$-certifiable training. We repeated training 4C3F networks 9 times for BCP and IBP models respectively, evaluated them, and reported the results in Table S2. We used the pre-trained 4C4F network for CAP as in the previous experiments. We set $\epsilon_{target}$ and $\epsilon_{eval}$ to $8/255$. For BCP and IBP, we present median, maximum, and minimum values of 9 results. Table S2 shows that BCP has comparable performance to IBP and outperforms CAP.

Table S2: Comparison to other $\ell_\infty$-verifiable training methods. Best performances are highlighted in bold.

| Method | Accuracy (%); median (max/min) | | |
|---|---|---|---|
| | standard | PGD | Verification |
| BCP | **36.50** (38.56/**34.65**) | **27.82** (29.53/**26.85**) | **24.20** (25.45/**22.01**) |
| IBP | 35.81 (**38.62**/31.23) | 27.30 (**30.20**/23.73) | 23.68 (**26.68**/21.18) |
| CAP | 19.00 | 17.33 | 16.06 |

Figure S3: The verification accuracy when varying the $\ell_2$-perturbations $\epsilon_{eval}$. We use $\epsilon'_{target} = 2$ on MNIST dataset (Top), and $\epsilon_{target}$ (Middle) and $2\epsilon_{target}$ (Bottom) on CIFAR-10 dataset for training, which are represented with the vertical lines.

**Sensitivity analysis on the schedule parameter $\lambda_0$ and $\lambda_1$**  We gradually increase $\lambda$ in (13) from $\lambda_0 = 0$ to $\lambda_1 = 1$ during training. In this section, we perform a sensitivity analysis on the parameters $\lambda_0$ and $\lambda_1$. We train a model with the 4C3F architecture on CIFAR-10. Table S3 shows that the performance has little to do with the initial weight $\lambda_0$ but it is highly related with the final weight $\lambda_1$. When the final weight $\lambda_1$ is close to 1, we get a lower standard accuracy and a higher verification accuracy. We trained three models for each pair $(\lambda_0, \lambda_1)$, and report average performance measures in Table S3.

Figure S4: Trade-off graph between standard accuracy and robustness. The vertical line segments indicate the verification accuracy (lower end points) and the PGD accuracy (upper end points).

Table S3: Sensitivity analysis on the parameters $\lambda_0$ and $\lambda_1$. We provide three performance measures (standard/PGD/verification accuracy (%)).

| | $\lambda_0$ | | | |
|---|---|---|---|---|
| $\lambda_1$ | **1** | **0.9** | **0.5** | **0.01** |
| **1** | 66.13/59.92/49.72 | 66.29/59.90/49.90 | 66.47/59.65/49.96 | 66.09/60.00/49.95 |
| **0.9** | | 67.18/60.52/48.99 | 66.90/60.23/49.19 | 66.91/60.50/49.41 |
| **0.5** | | | 69.86/61.08/42.32 | 69.96/61.19/42.70 |
| **0.01** | | | | 71.20/59.57/20.69 |

Figure S5: Illustration of the outer bounds for the BCP trained model on ImageNet.

## Footnotes

[1]`https://github.com/locuslab/convex_adversarial/model_scaled_l2`