[Reviews · NeurIPS 2020]

Review 1

Summary and Contributions: The main contribution of this paper is to propose a certifiable training method with a tight outer bound propagation in a highly efficient manner. To my best of knowledge, this is the first non-trivial determinist verification procedure/method that can be applied on Tiny ImageNet.

Strengths: Overall, the paper is well-written and easy to follow the logic. The idea is simple and yet effective in practice, which is well motivated by the geometric interpretation, i.e., get a tighter outer bound (image space). I really like Figure 1. All the technical proof and induction is correct and sound. Extensive experiments are conducted to support their proposed methods.

Weaknesses: It would be great if you can give a simple example like figure 2 to showcase the gap between the proposed method and other baselines (dual relaxation-based, SDP). A tighter or looser region you developed? or do you sacrifice something? the trade-off between efficiency and accuracy?

Correctness: Yes.

Clarity: Yes.

Relation to Prior Work: Yes.

Reproducibility: Yes

Additional Feedback: I just have some minor comments. 1. For the problem (11), it can be recast into a one-dimensional search problem, (e.g., see the associated lagrangian multiplier w.r.t ball constraints). Naturally, it can be done by some modified secant method to solve it. I am not sure which one is better. Maybe you can have a try. 2. prop 1, why do you cite [39]? 3. line 125, 0? 4. line 216, as you mentioned figure 2(d) at first, it's better to change it to (a). #Update: Thanks for your response.


Review 2

Summary and Contributions: This paper proposed a method named Boxed constraint propagation (BCP) for both verification and certified robust training. The key idea is to simply combine interval bound propagation as the Linf box constraint and additional L2 ball constraint whose radius is estimated by Lipschitz constant (it appears to me that the lipschitz constant here is global lipschitz constant since the authors use power iterations to get rho. The authors didn't explain clearly, but it appears to me that they use power iteration to get maximum eigen-value of weight matrix which should correspond to global lips const when input perturbation is L2 norm. Please clarify if this is not what you are doing in the paper). The results compared with Wong etal, which is not state-of-the-art in the certified defense method; meanwhile, the improvement of BCP is marginal. Also, the evaluation on the tightness of the bounds is problematic: removing the box constraint would naturally lead to loose bound due to the way that the authors estimate rho. I think a reasonable comparison would be comparing BCP (i.e. equation 11) with IBP on L2 input perturbation and see if the tightness is improved. Overall, the novelty is low and the performance of the method is marginal.

Strengths: -

Weaknesses: 1. Evaluation is problematic and does not compare to state-of-the-art in the certified training (which should be at least IBP, Gowal 2019), please see the "summary and contributions" part for more details. 2. Novelty is relatively low and the improved performance is marginal

Correctness: Evaluation is problematic

Clarity: The notation is a bit dense

Relation to Prior Work: It discussed some certified training methods

Reproducibility: Yes

Additional Feedback: Other questions: 1. Line 125: what does "0 method" mean? 2. Line 204: LMT is not STOA, should compare at least with IBP 3. Fig 2: not clear what it is plotting. Why w/o BCP goes outside the decision boundary? 4. Tab 1: improvement is marginal, and Wong is not STOA ===== post rebuttal ====== The additional experiments compared with IBP on L2 norm shows large improvement. Please include this experiment in the revised version. Also, it is suggested to add all the clarifications as well as simplifying the math derivation in the revision to improve the manuscript.


Review 3

Summary and Contributions: The paper proposes a certifiable training algorithm based on Lipschitz analysis and interval arithmetic. They show that BCP achieves a tighter outer bound than the global Lipschitz-based outer bound. The computation time has been improved.

Strengths: The paper is well-organized. The authors reasonably explain their motivation and methodology. The significance and novelty are median.

Weaknesses: I have concerns for Algorithm 1 and find it not easy to follow. Particularly to the $\rho^{K-1}$, computed with product sequence. Also, c = W_1-W_0, while z is a vector as described in the notation. How can they have the same dimension for substraction p = z-\\rho^{K-1} c/||c||? In the experiments, the paper stated inline 157 that their methods using the l_infity case could be considered as IBP (interval bound propagation). Although they have compared with IBP given in the appendix, I wonder why they did not put it to the main text.

Correctness: The value $\rho^{K-1}$ could be significantly small or large. I have checked the code, which they set " r = eps*torch.ones(x.size()[0]).cuda() # b, if args.linfty: r *= np.sqrt(x.reshape(x.size(0),-1).size(1))" They set linfty=False meaning that by default one should not use this estimation (the description of the algorithm should be consistent with what is implemented ). Perhaps the author could use normalization methods (batch normalization/ layer normalization) to stabilize the Lipchitz constant. I think the author should give more descriptions in their algorithm. It's not easy to follow and difficult to find the right hyper-parameters. ============================== I have read their rebuttal, which clarifies my concerns and question. Althought the clarity need to be improved, I am willing to increase the score to borderline accept. I would suggest the author make the algorithm in more details without referring to other paragraphs.

Clarity: Median

Relation to Prior Work: Median

Reproducibility: No

Additional Feedback:

[Author Response · NeurIPS 2020]

We thank the reviewers for the valuable feedback. We will reflect minor errors instantly and try suggestions in the
future. We want to address concerns and to clarify misconceptions. In this rebuttal, CAP denotes Wong et al.
**[R2 R3] Under $\ell_2$-norm, IBP does not work well enough.** In IBP, the authors didn't mention the certifiable training
for $\ell_2$-norm. In CROWN-IBP, however, the authors explained that IBP can be applied to $\ell_2$-norm. We run the code for
IBP provided by the authors of CROWN-IBP on CIFAR-10 under $\ell_2$-setting with $\ell_2$-norm $36/255$, where we considered a
wide range of parameters including those suggested by CROWN-IBP and our range of settings described in Section C.
IBP achieved the verification (standard) accuracy of 22.6-23.0% (31.3-33.5%) which was inferior to CAP = 50.29%
(60.14%) and to LMT = 37.20% (56.49%). It also implies LMT-bound is much tighter than IBP-bound under $\ell_2$-norm.
Thus, we compared our method to LMT and CAP rather than IBP under $\ell_2$-case in the main text. **[R2 R3] On the**
**novelty.** BCP was carefully designed in a layer-wise manner (Fig1) to obtain the tighter outer bound. One may simply
use IBP to get the additional box constraint in (11), but this box constraint is redundant because it is much looser
than the $\ell_2$-constraint in (11). BCP is designed to provide a nonredundant box constraint in (11) to tighten the bound.
We observed the tightness in Fig2 and Fig3. As a result, BCP outperforms both LMT and IBP in a large margin
(>12-28%p) under $\ell_2$-norm. For further intuition behind the layerwise design of BCP, refer to Section B.1. **[R2 R3]**
**BCP outperforms the others with a meaningful margin.** In Tab 1, the evaluation results at a single $\epsilon_{eval}$ seem to be
a marginal improvement compared to CAP. However, when considering a wide range of $\epsilon_{eval}$ in Fig4 (and in FigS3),
BCP outperforms CAP by 3.7-5.6%p in standard accuracy on CIFAR-10. For $\epsilon_{eval} > 36/255$, BCP outperforms CAP in a
large margin. For example, when evaluating at $72/255$, BCP (34.2%) defeats CAP (23.9%) by 10.3%p. Moreover, only
BCP can achieve a meaningful verification accuracy on Tiny ImageNet, while others cannot. **[R1 R2] Fig2 illustrates**
**how BCP can tighten the outer region by introducing the box constraint.** In Fig2 (a)-(c), we can easily visualize
the high-dimensional ellipsoid $h_K(\mathbb{B}_2^{(K-1)}) \subset \mathbb{R}^c$ in 2D plane with $\zeta_y$- and $\zeta_{m'}$-axes (line 211-212) by projection.
However, a high-dimensional parallelogram $h_K(\mathbb{B}_\infty^{(K-1)})$ for $c > 2$ is hard to visualize in the 2D plane. Thus, we
use the red lines in Fig2 to indicate that the projection of the outer region $h_K(\mathbb{B}_2^{(K-1)} \cap \mathbb{B}_\infty^{(K-1)})$ must lie above the
red line and inside the ellipsoid. Based on (9), we used the verification boundary ($\zeta_y - \zeta_{m'} \geq \zeta_y^* - \zeta_{m'}^*$), where
the red line is obtained from the solution $\zeta_y^*, \zeta_{m'}^*$ of (9) for $\hat{z}(\mathbb{B}(\boldsymbol{x})) = h_K(\mathbb{B}_2^{(K-1)} \cap \mathbb{B}_\infty^{(K-1)})$, and the blue line is
for $\hat{z}(\mathbb{B}(\boldsymbol{x})) = h_K(\mathbb{B}_2^{(K-1)})$. Fig2 (d) explicitly illustrates the ellipsoid and the parallelogram with the verification
boundary for a toy binary classification ($c = 2$). Fig2 shows typical examples for which the verification succeeds with
BCP but fails without BCP (R2-8-3). On comparing the gap between BCP and other baselines, the visualization of the
logit space can be inappropriate because it is not possible to directly compare the logits from the models obtained by
different certifiable training methods. As verification methods, comparing only verification phase using one trained
model can be unfair because the performance highly depends on which method the model is trained with. **[R2 R3] We**
**computed the radius $\rho^{(k)}$ with the layerwise Lipschitz constants $L^{(i)}$'s.** We computed $L^{(i)}$ for each $i$-th operation
(BCP.py line 254,286), and then computed $\rho^{(k)} = \epsilon \Pi_{i=1}^k L^{(i)}$, multiplying the expansion rate (=Lipshitz constant)
through each layer (line 114-118,137-139, Section B.2). We are sorry to make R2 confused. Lipschitz constant usually
refer to the global Lipschitz constant (gL) rather than a local one (lL) when not specified. We used the gL for the
efficient computation (line 127-128). For the layerwise Lipschitz constant, when the layerwise operation $h_k$ is linear, the
maximum eigen-value of weight matrix corresponds to both gL and lL (gL = lL). **[R3] On the stabilization of $\rho^{(K-1)}$.**
In FigS1 (top), the Lipschitz constant keeps increasing through standard training. To stabilize the Lipschitz constant,
we applied a commonly-used scheduling scheme on $\epsilon$ and $\lambda$ during the BCP training (line 195-200). In FigS1 (bottom),
$L^{(-1)} = \Pi_{i=1}^{K-1} L^{(i)}$ is about 10 after training with BCP, and it is multiplied by $\epsilon$ to provide $\rho^{(K-1)} = \epsilon L^{(-1)}$. Thus,
$\rho^{(K-1)}$ is about $10\epsilon$. In the code, we initialize $\rho^{(0)} = \epsilon$ (BCP.py line 214) and update the radius by $\rho^{(i+1)} = \rho^{(i)} * L^{(i+1)}$
(`r = r*p`) for each layer as implemented in BCP.py line 236-243,270,300. We also tried BN, but it is not effective
to improve robustness. Moreover, there is a paper named "Batch Normalization is a Cause of Adversarial Vulnerability".
**[R3] On the implementation of BCP.** Our main focus is on $\ell_2$-certifiable training. In the main text, we describe the
BCP algorithm and provide the results for $\ell_2$-case, so, in the code, we set `args.linfty=False` as default. However,
BCP can be applied to $\ell_p$-norm for any $p \in (0, \infty]$ (line 154-155). In appendix, we presented the results for $\ell_\infty$-norm
in Tab S2, and the implementation is available by setting `args.linfty=True` in the code (BCP.py line 216-217).
Therefore, the description of the algorithm, including the $\ell_\infty$-case (line 154-159), is consistent with the implementation.
Moreover, for reproducibility, we provided how to run the code in one-line command in README.md and explained
details of the hyper-parameters in Section C, including the scheduling on $\epsilon$ and $\lambda$. For reference, in the $\ell_\infty$-case, BCP is
not IBP but a generalized version of IBP because BCP still uses the $\ell_2$-bound propagation. **[R3] c and $\mathbf{z}^{(K-1)}$ have the**
**same shape.** The notation $\mathbf{W}_i$ is for the $i$-th row vector of the matrix $\mathbf{W}$ (line 144). Let $\mathbf{W}^{(K)} \in \mathbb{R}^{m \times n}$, then $\mathbf{W}_i^{(K)}$
is an $n$-dimensional row vector, and $\mathbf{z}^{(K-1)}$ is an $n$-dimensional column vector. Therefore, since $\mathbf{c}^T = \mathbf{W}_1^{(K)} - \mathbf{W}_0^{(K)}$,
$\mathbf{c}$ has the same dimension to $\mathbf{z}^{(K-1)}$. **[R1] Trade-off between efficiency and accuracy.** BCP is much faster than CAP
(Tab1) as well as achieving the better performance in a wide range of $\epsilon_{val}$ (Fig4). BCP can outperform LMT and IBP in
a large margin with affordable computational overhead.

[Meta-Review · NeurIPS 2020]

The paper proposes a certified robust training method based on Lipschitz bounds. The reviewers initially have concerns due to lack of clarity and problematic experimental comparisons. Those concerns were addressed by the rebuttal and all the reviewers agreed to weakly accept the paper since it outperforms existing methods on L2 certified robust training. We hope the authors can improve clarity and carefully incorporate the rebuttal into the main paper.